# Genomic Instability of Circulating Tumor DNA as a Prognostic Marker for Pancreatic Cancer Survival: A Prospective Cohort Study

**DOI:** 10.3390/cancers13215466

**Published:** 2021-10-30

**Authors:** Sang Myung Woo, Min Kyeong Kim, Boram Park, Eun-Hae Cho, Tae-Rim Lee, Chang-Seok Ki, Kyong-Ah Yoon, Yun-Hee Kim, Wonyoung Choi, Do Yei Kim, Jin-Hyeok Hwang, Jae Hee Cho, Sung-Sik Han, Woo Jin Lee, Sang-Jae Park, Sun-Young Kong

**Affiliations:** 1Center for Liver and Pancreatobiliary Cancer, National Cancer Center, Goyang 10408, Korea; wsm@ncc.re.kr (S.M.W.); sshan@ncc.re.kr (S.-S.H.); lwj@ncc.re.kr (W.J.L.); spark@ncc.re.kr (S.-J.P.); 2Cancer Biomedical Science, National Cancer Center Graduate School of Cancer Science and Policy, Goyang 10408, Korea; sensia37@ncc.re.kr (Y.-H.K.); wonyoungchoi@ncc.re.kr (W.C.); 3Division of Translational Science, National Cancer Center, Goyang 10408, Korea; 72952@ncc.re.kr; 4Biostatistics Collaboration Team, Research Core Center, Research Institute, National Cancer Center, Goyang 10408, Korea; brknow@naver.com; 5Genome Research Center, GC Genome, Yongin 16924, Korea; ehcho@greencross.com (E.-H.C.); leetaerim@greencross.com (T.-R.L.); cski@greencross.com (C.-S.K.); doyei@greencross.com (D.Y.K.); 6College of Veterinary Medicine, Konkuk University, Seoul 05029, Korea; kayoon@konkuk.ac.kr; 7Molecular Imaging Branch, Division of Convergence Technology, National Cancer Center, Goyang 10408, Korea; 8Department of Internal Medicine, Seoul National University College of Medicine, Seoul National University Bundang Hospital, Seongnam 13620, Korea; jhhwang@snubh.org; 9Department of Internal Medicine, Gachon University Gil Medical Center, Incheon 21556, Korea; jjhcho9328@gmail.com; 10Department of Internal Medicine, Gangnam Severance Hospital, Yonsei University College of Medicine, Seoul 06273, Korea; 11Department of Laboratory Medicine, National Cancer Center, Goyang 10408, Korea

**Keywords:** pancreatic ductal adenocarcinoma, genomic instability, prognostic biomarker, outcome prediction, instability score, circulating tumor DNA, pancreatic cancer, genomic instability index

## Abstract

**Simple Summary:**

This prospective cohort study showed that circulating tumor DNA-genomic instability (ctDNA-GI) I-scores, which was defined as the natural logarithm of the sum of LOESS-normalized Z-scores of sequenced reads in 1 Mb bins, are prognostic of the outcome of either localized or metastatic pancreatic adenocarcinoma. At baseline, 24.1% of patients had high genomic instability with I-score. Multivariable analyses demonstrated I-score was a significant factor for progression-free survival and overall survival.

**Abstract:**

Genomic instability of circulating tumor DNA (ctDNA) as a prognostic biomarker has not been evaluated in pancreatic cancer. We investigated the role of the genomic instability index of ctDNA in pancreatic ductal adenocarcinoma (PDAC). We prospectively enrolled 315 patients newly diagnosed with resectable (*n* = 110), locally advanced (*n* = 78), and metastatic (*n* = 127) PDAC from March 2015 through January 2020. Low-depth whole-genome cell-free DNA sequencing identified genome-wide copy number alterations using instability score (I-score) to reflect genome-wide instability. Plasma cell-free and matched tumor tissue DNA from 15 patients with resectable pancreatic cancer was sequenced to assess the concordance of chromosomal copy number alteration profiles. Associations of I-score with clinical factors or survival were assessed. Seventy-six patients had high genomic instability with I-score > 7.3 in pre-treatment ctDNA; proportions of high I-score were 5.5%, 5.1%, and 52% in resectable, locally advanced, and metastatic stages, respectively. Correlation coefficients between Z-scores of plasma and tissue DNA at segment resolution were high (r^2^ = 0.82). Univariable analysis showed the association of I-score with progression-free survival in each stage. Multivariable analyses demonstrated that clinical stage-adjusted I-scores were significant factors for progression-free and overall survival. In these patients, ctDNA genomic I-scores provided prognostic information relevant to progression-free survival in each clinical stage.

## 1. Introduction

Pancreatic ductal adenocarcinoma (PDAC) is a public health problem because of its dismal prognosis and increasing incidence. Despite the improvements in diagnosis and therapy applied during the past few decades, the five-year survival rate for pancreatic cancer is 10% worldwide [1,2]. Treatment options depend on several factors, including the cancer type and stage, possible side effects, and patient preferences and overall health conditions. Thus, a better understanding of the biology pertinent to PDAC might lead to more efficacious therapeutic strategies.

Genomic instability is a typical hallmark of cancer. It promotes inter- and intra-tumor heterogeneity and enables cancer cell adaptation to environmental stress, thereby driving aggressive tumor behavior and resistance to cancer therapies [3,4]. Moreover, recently integrated whole-genome analysis uncovered that the molecular subtypes of pancreatic cancer are linked to specific copy number aberrations in genes such as mutant *KRAS* and *GATA6* [5]. These data support the premise that the constellation of genomic aberrations in the tumor gives rise to the molecular subtype associated with disease progression.

Liquid biopsies are of particular interest from a clinical perspective because they are non-invasive and assess biomarkers released by primary tumors and metastases that reflect tumor biology [6]. Most studies have assayed *KRAS* oncogene mutations to identify circulating tumor DNA (ctDNA), and several groups, including ours, have reported that the presence of a *KRAS* mutation has a negative influence on the prognosis of pancreatic cancer patients [7,8,9,10]. However, studies for genomic instability in ctDNA have not been thoroughly conducted.

Chromosomal structural variations such as chromosomal rearrangement, duplica-tion, and deletion are prominent mechanisms of genomic damage in pancreatic cancer [11,12]. Surrogate measures of defects in DNA maintenance have potential therapeutic selection implications. Irinotecan and nanoliposome-encapsulated irinotecan, combined with 5-fluorouracil and/or oxaliplatin, have become the main cytotoxic agents for pancreatic cancer. Defects in DNA repair and DNA damage checkpoints have been identified with enhanced sensitivity to topoisomerase 1 inhibitors [13]. Furthermore, the combination of tumor copy number alterations and mutation load suggested a better predictor for identifying patients most likely to respond to immunotherapy than the mutation burden alone [14,15]. Current studies have elucidated that poly ADP ribose polymerase (PARP) inhibitors and platinum agents might be effective for inducing tumor regression in solid tumors bearing an unstable genome, including pancreatic cancer [11]. 

However, there are significant hurdles to overcome as technical challenges of DNA sequencing using small diagnostic samples preserved in fixatives such as formalin, analytical demands, and the return of results within a clinically relevant timeframe [11]. Therefore, identifying genomic instability in ctDNA can predict outcomes more effectively and increase the efficacy of treatment chemotherapy. Here, we investigated genomic instability in ctDNA as a prognostic and predictive marker of survival and the therapeutic response in PDAC.

## 2. Materials and Methods

### 2.1. Study Design and Sample Collection

This study prospectively enrolled 315 patients newly diagnosed with PDAC between March 2015 and January 2020. Patient blood samples and clinical data were collected from three hospitals: The National Cancer Center; the Seoul National University Bundang Hospital; and the Gachon University Gil Medical Center, Republic of Korea. This study was approved by Institutional Review Board (IRB No. NCC2015-0054, NCC2016-001, and NCC2019-027), and the participating patients gave their informed consent. Patients were divided into three clinical-stage groups: patients with (1) surgical resection, (2) local but unresectable disease (locally advanced), and (3) metastatic disease. Blood samples were collected before and after treatment, every three months for resectable and locally advanced patients, and every two months for metastatic patients, with restaging imaging after initiation of anticancer treatment. We included patients who received FOLFIRINOX (FOL-folinic acid, F-fluorouracil, IRIN-irinotecan, OX-oxaliplatin) therapy as first-line treatment, with the aim to explore the application of genomic instability in ctDNA in monitoring tumor burden change following treatment. Treatment response was assessed every four cycles of chemotherapy via abdominal enhanced computerized tomography (CT) and/or enhanced magnetic resonance imaging (MRI) of the liver. According to the Response Evaluation Criteria in Solid Tumors (RECIST) version 1.1. [16], tumor response was quantitatively defined as complete response (CR), partial response (PR), stable disease (SD), and progressive disease (PD). 

Formalin-fixed paraffin-embedded (FFPE) tissues (*n* = 15) were obtained from the biobank of the National Cancer Center, Korea. Thirty-eight healthy controls were enrolled from the National Cancer Center (IRB No. NCC2017-0083).

### 2.2. Sample Processing and DNA Extraction

Up to 10 mL of peripheral blood was collected by venipuncture in two types of evacuated blood collection tubes: K2EDTA (BD #366643; Becton Dickinson and Company, Franklin Lakes, NJ, USA) or Cell-Free DNA Streck BCT (Streck #218962; Streck, Omaha, NE, USA). Plasma from blood collected in K2EDTA tubes was separated within two hours after drawing blood to ensure ctDNA integrity. Processing of blood from Cell-Free DNA Streck BCT tubes was carried out within five days. Whole blood was centrifuged at 1600× *g* for 10 min, after which the supernatant was centrifuged again at 16,000× *g* for 10 min to remove any remaining contaminating cells. Supernatants were immediately stored at −80 °C until use.

Plasma ctDNA was extracted from 1–2 mL of plasma using a QIAamp Circulating Nucleic Acid Kit (Qiagen Cat# 61504, Hilden, Germany) or Chemagic cfDNA 2 K kit (PerkinElmer, Waltham, MA, USA). The final DNA eluent (50 μL) was quantified by Qubit 2.0 Fluorometer with a Qubit dsDNA HS (High Sensitivity) assay kit (Cat# Q32851, Life Technologies, Carlsbad, CA, USA). Genomic DNA from FFPE tissues was extracted using a GeneRead FFPE DNA Kit (Cat# 180134, Qiagen).

### 2.3. Library Preparation for Whole-Genome Sequencing of Cell-Free DNA

A Tapestation 4200 (Agilent Technologies, Santa Clara, CA, USA) was used to examine the size of cell-free DNA (cfDNA) fragments before library construction, and samples showing a proper size distribution were used for library construction (Appendix A). DNA libraries were prepared using a TruSeqNano kit (Illumina Inc., Cat# FC-121-4003, San Diego, CA, USA). Briefly, approximately 5 ng of cell-free DNA (cfDNA) was subjected to end-repair, adenylation, and adaptor ligation. The pooled libraries of 28 samples per run were analyzed with a NextSeq 500 instrument (Illumina Inc.) using 75 bp single-end read mode.

### 2.4. Whole-Genome Sequencing of Tumor Tissue DNA

The extracted genomic DNA was sheared to 180–250 bp using an M220 focused ultrasonicator (Covaris, Woburn, MA, USA). The sheared genomic DNA sizes were verified with a Tapestation4200. Libraries were constructed using an Accel-NGS 2S plus DNA Library Kit (Swift Biosciences Inc., Ann Arbor, MI, USA). We sequenced an average of 3.1 million reads on a NextSeq 500 system (Illumina, San Diego, CA, USA) using 75 bp single-end read mode.

### 2.5. Genomic Instability Calculation (I-Score)

Sequenced reads were aligned to the hg19 human reference genome using the BWA-mem algorithm (0.7.5.a) with default parameters [17]. PCR duplicates were removed with Picard release 1.96 (https://broadinstitute.github.io/picard/, accessed on 23 August 2014), and reads with mapping quality below 60 were excluded from further analysis. After splitting the whole genome into 2897 1 Mb bins, 163 bins located in low mapping regions, such as telomeres, were not used for genomic instability calculation. The relative frequency of sequencing reads mapped to each bin was calculated and corrected for GC content bias using the LOESS algorithm [18]. To measure the local instability in patients with PDAC, the Z-score was calculated by comparing the relative frequency for each bin with that of 38 healthy control subjects. A Z-score for each bin was calculated with the formula below (Equation (1), Figure 1a):(1)$$Z-{\mathrm{score}}_{\mathrm{bin}}=\frac{{\mathrm{RF}}_{\mathrm{bin}}-{\;M}_{\mathrm{bin}}\;}{{\mathrm{SD}}_{\mathrm{bin}}}$$
where RF_bin_ is the relative frequency of a bin in a patient with PDAC, M_bin_ is the mean of relative frequencies of a bin in normal healthy subjects, and SD_bin_ is the standard deviation of relative frequencies of a bin in normal healthy subjects.

To measure the extent of genome-wide copy number instability, we developed an I-score that summarizes the local Z-scores into a single value. High I-score means a high level of genomic instability. LOESS algorithm was applied to smooth Z-scores of adjacent bins before I-score calculation, which helps reduce noise. I-score was calculated as described below Equation (2):(2)$$I-\mathrm{score}=\ln\sum |{\mathrm{LOESS}\;\mathrm{smoothed}\;Z}_{\mathrm{bin}}|$$

### 2.6. Identification of Recurrent Copy Number Alterations Regions in Pancreatic Cancer Patients

Hierarchical clustering analysis was performed using the heatmap.2 function in the R software package gplots [19]. Genome segmentation analysis was carried out using the DNA copy R software package [20] to divide the genome into equal DNA copy number regions, which are called copy number segments. The genomic regions recurrently amplified or deleted across the 315 patients with PDAC were identified using the Genomic Identification of Significant Targets in Cancer (GISTIC2.0) algorithm [21]. The cutoff for statistical significance was set to a false-discovery rate (FDR) adjusted *p*-value (q-value) < 0.25. Default parameters were used for GISTIC analysis.

### 2.7. Gene Sum Score Calculation and Validation

The genomic regions identified by GISTIC analysis were divided into gene-level intervals, and Z-score for each gene was calculated. Each gene was scored 1 or 0 according to its Z-score. Genes in GISTIC amplification regions with a Z-score higher than 2 or genes in GISTIC deletion regions with a Z-score less than −2 were scored as 1. All other genes were scored as 0. To select the minimum subset of genes having a prognostic impact, we tested the overall survival (OS) difference between patients with gene scores 1 and 0 for all GISTIC genes using log-rank tests. Five-fold cross-validation within 315 patients with PDAC was carried out for internal validation. In each of the cross-validation training sets, genes were sorted by the log-rank *p*-values and the top N most significant genes were independently selected in one of the five CV training sets, and the gene sum score (GSS) was defined as the sum of scores of selected genes in each CV training set. Top N genes ranging from 1 to 50 were investigated and the best combination of top N genes and the cutoff value were finally set in each cross-validation training set. The optimal cutoff values for GSSs were set to the point with the most significant log-rank test split. Further external validation was conducted using a publicly available data set that analyzed putative gene-level copy number profiles from tumor tissue DNA in patients with PDAC [22,23]. Genes labeled as high-level amplification or homozygous deletion were scored to 1 for GSS calculation in the external data set. Five GSSs were calculated, and the prognostic impacts of GSS on OS were validated. Schematic illustration of the GSS calculation and validation workflow can be found in Appendix A.

### 2.8. Comparing Copy Number Aberration Profiles of ctDNA and Matched Tumor Tissue DNA

Plasma ctDNA and matched tumor tissue DNA were sequenced from fifteen resectable pancreatic cancer patients to assess the degree of concordance between chromosomal copy number alterations (CNA) profiles. The similarity between CNA profiles was measured with Pearson’s correlation coefficients comparing Z-scores at segment resolution. Genomic regions segmented with tumor tissue DNA were assumed as the baseline, and the Z-score for each plasma DNA segment was calculated as the mean value of Z-scores of 1-Mb size bins covering the baseline regions. To calculate Z-scores for tissue DNA, each bin reference value was constructed using 2012 samples from healthy females.

### 2.9. Statistical Analysis

Associations between I-score and other clinical factors were tested with two-sample *t*-tests, Pearson chi-square tests, or Fisher’s exact test for each variable. We aimed to test the hypothesis that the detection of the I-score is associated with clinical outcome. OS and progression-free survival (PFS) were the primary outcomes. OS was calculated from the day of diagnosis to the day of last follow-up or death from any cause. PFS was measured from the day of diagnosis to the day of progression or death. The associations of I-score with PFS and OS were assessed using univariable or multivariable Cox proportional hazards models. The effects were presented as hazard ratio (HR) and 95% confidence interval (CI). After backward variable selection with an elimination criterion of *p* < 0.05, only the stage was adjusted in PFS and OS multivariable models. Survival curves were estimated with the Kaplan–Meier method, and the survival difference was tested using a log-rank test. The ability of factors to predict PFS and OS was assessed with Harrell’s C-Index. All patients were categorized into binary I-score groups (low and high) using the method proposed by Contal and O’Quigley [24], based on the log-rank test statistic. A two-sided *p*-value < 0.05 was considered statistically significant. All statistical analyses were performed with SAS (version 9.4; SAS Institute Inc., Cary, NC, USA) and R statistical software (version 3.6.2, R Foundation for Statistical Computing, Vienna, Ausrtria).

## 3. Results

### 3.1. Patient Demographics and Distribution of I-Score

The clinical details, including age, sex, performance status, and tumor status, are provided in Table 1. Of the 315 patients enrolled, 181 were men, the mean age was 65.4 ± 9.7 years, and the median follow-up period was 18.9 months (range, 0–55.6 months). Patients with resectable, locally advanced, and metastatic cancers accounted for 34.9% (*n* = 110), 24.8% (*n* = 78), and 40.3% (*n* = 127) of all patients, respectively (Appendix A).

We employed an I-score cutoff value of 7.3 to divide patients into low and high groups, as determined by Contal and O’Quigley [24]. Among 315 patients, 76 (24.1%) had pre-treatment ctDNA I-scores higher than the cutoff values (Table 1). Age, sex, and Eastern Cooperative Oncology Group (ECOG) performance status were not significantly different between the low and high I-score groups. The rate of high I-score in ctDNA was higher in the metastatic group (*n* = 66, 52.0%) than in resectable (*n* = 6, 5.5%) or locally advanced (*n* = 4, 5.1%) groups. The frequency of high carbohydrate antigen 19-9 (CA19-9) and carcinoembryonic antigen (CEA) levels was 31.2% and 42.1%, respectively, were higher in the high I-score group than the low I-score group. The I-score has a significant association with clinical stage, tumor location, CA19-9 level, and CEA level (*p* < 0.001).

### 3.2. Prognostic Impact of ctDNA I-Score in Pancreatic Cancer

In the univariable model, the high I-score group had a significantly higher HR than that of the low I-score group in PFS (HR, 2.69, 95% CI, 1.96–3.69; *p* < 0.001) and OS (HR, 3.07, 95% CI, 2.21–4.25; *p* < 0.001) (Table 2 and Figure 2). In addition, clinical stage (HR, 2.76; 95% CI, 1.99–3.84, and HR, 4.05; 95% CI, 2.58–5.82), CA19-9 level (HR, 1.69; 95% CI, 1.22–2.33, and HR, 1.60; 95% CI, 1.14–2.24), and CEA concentration (HR, 1.74; 95% CI, 1.28–2.35, and HR, 2.09; 95% CI, 1.51–2.88) were significant factors for PFS as well as OS. In the stage-adjusted multivariable model, the I-score was still significantly associated with PFS (HR, 1.99; 95% CI, 1.42–2.77; *p* < 0.001) and OS (HR, 2.15; 95% CI, 1.53–3.01; *p* < 0.001). Subsequent subgroup analysis of survival showed that significant differences between the high I-score group and the low I-score group with PFS were observed in patients with resectable disease (HR, 2.61; 95% CI, 1.11–6.13; *p =* 0.0276), locally advanced (HR, 5.90; 95% CI, 1.70–20.51; *p =* 0.0053), or metastatic cancer (HR, 2.05; 95% CI, 1.28–3.29; *p =* 0.0028). There were significant differences in OS rates between the high and low groups in patients with metastatic disease (HR, 2.36; 95% CI, 1.47–3.80; *p* < 0.001), but not in patients with resectable or locally advanced diseases (Appendix A).

### 3.3. ctDNA I-Score Is Associated with Response to Chemotherapy

The responses of 48 patients receiving FOLFIRINOX as first-line chemotherapy are summarized in Appendix A. The objective response rate was 46.7% in the high I-score group and 17.6% in the low I-score group (*p* = 0.076). Figure 3a shows the responses according to changes in I-score in 18 metastatic patients at 3 months. At 6 months, two patients with 37.2% and 9.6% increased I-score levels showed PD (Figure 3b). In contrast, two (40%) of five patients with a 13.1% or 15.9% decrease in I-score levels showed PD. Among those with decreased scores, one with a decrease of 25.1% had a PR, and two with a decrease of 26.7% and 32.3% had SD. A representative ctDNA I-score plot demonstrates a dynamic range within an individual patient whose liver metastasis had initially responded and progressed during the administration of gemcitabine-based chemotherapy, which was associated with an initial drop and subsequent increase in ctDNA I-score (Figure 3c).

### 3.4. Identifying Recurrent CNAs in Pancreatic Cancer

According to their I-score, the unsupervised hierarchical clustering analysis showed that the 315 patients were clustered into two distinct groups (Figure 1c). All 60 patients in the first group had high I-scores ranging from 7.4 to 10.1, showing recurrent amplification and deletion patterns. In contrast, the second group consisted of 255 patients with relatively low I-scores, and only 16 patients had an I-score higher than 7.3. GISTIC analysis identified five amplification and three deletion regions statistically significant across 315 patients (Figure 1b, Appendix A). GISTIC amplification regions included 1698 genes, and 36 were reported to be frequently, more than 10% of the study cohort, amplified in pancreatic adenocarcinoma patients. Similarly, GISTIC deletion regions included 364 genes and 22, which have been frequently deleted (Appendix A). We found that these GISTIC regions overlapped cancer-related genes as well, such as oncogenes, tumor suppressor genes, and genes related to poor prognosis when amplified or deleted [25,26,27,28,29]. Among the five amplification regions, four overlapped with seven oncogenes and five candidate genes. Similarly, one deletion region contained four tumor suppressor genes and one oncogene (Table 3).

Clear CNAs were identified in all tumor tissues from 15 patients. One patient showed a detectable copy number-gain and loss pattern in plasma ctDNA. It is noteworthy that the profile of CNAs observed in plasma is highly similar to CNAs in tissue (Appendix A). Correlation coefficients between Z-scores of plasma ctDNA and tissue DNA at segment resolution reached 0.82 (Appendix A).

### 3.5. GISTIC Genes and GSS

In each cross-validation set, the top N most significant genes ranging from 14 to 43 were selected to calculate GSSs. Then, the optimal cutoff values for each GSS were set to the point that maximizes the OS difference between GSS-high and -low groups in cross-validation training sets (Appendix A). The prognostic impact of each GSS was internally validated in the matched cross-validation test set. There were significant (*p* < 0.05) differences in OS rates in three of the five cross-validation sets (Appendix A). In addition, GSS-high groups had a higher HR than the GSS-low groups in OS (Appendix A). GSS_Overlap and GSS_Union were calculated in the external data set using 8 genes repeatedly selected across all five cross-validation sets and 79 genes selected at least once from all cross-validation sets, respectively (Appendix A). By applying the cutoff value of 1 and 8, both GSS_Overlap and GSS_Union showed a significant prognostic impact in OS (Appendix A).

## 4. Discussion

This study of 315 patients is the most extensive prospective study evaluating the role of plasma ctDNA in PDAC. Although the strategy of using *KRAS* mutations in ctDNA as a tumor marker may be theoretically optimal in a disease like PDAC where *KRAS* mutation rates exceed 90%, the stochastic nature of circulating ctDNAs may lead to underestimation of the true circulating tumor burden or nature if detection is limited to a single mutation [9]. In the present study, ctDNA I-scores were prognostic of the outcome and predictive of response to systemic chemotherapy. Although patients with PDAC with a high I-score are highly responsive to chemotherapy, progressions are commonly observed within months despite treatment. We found that ctDNA I-score demonstrated a strong correlation with PFS or OS across different disease stages. Moreover, ctDNA I-scores may provide unique predictive information on chemotherapy outcome in localized disease and the metastatic setting. In contrast to the challenges of repetitive tissue biopsies, serial ctDNA I-score monitoring may provide an attractive alternative strategy for monitoring drug resistance and guiding treatment.

To the best of our knowledge, this study is the first to suggest the predictive value of ctDNA in patients with PDAC. In 49 patients with advanced PDAC receiving FOLFIRINOX, radiologic responses were significantly associated with ctDNA I-score. However, ctDNA I-scores were not able to differentiate patients with SD from patients with PD. Further, there were no significant differences between high and low I-score groups in terms of response duration, PFS, and OS. Although it did not reach statistical significance, changes in total ctDNA I-score represent a potential marker to predict treatment response. Additional large prospective studies are needed to investigate whether ctDNA I-scores can predict treatment response, considering that total cfDNA is not reliable for treatment response prediction due to low specificity for the overall tumor burden [30]. 

Comparison of the matched plasma and tissue DNA showed that the chromosomal CNA profile obtained from plasma cfDNA closely reflected CNAs in a tumor tissue, which implies the potential ability of cfDNA as a minimally invasive surrogate marker for tumor tissue DNA. Even though abnormal plasma CNAs were detected in only one of 15 patients, this low detection rate was due to the nature of resectable PDAC [31]. Considering that even *KRAS* mutations—one of the most common oncogenic variants in pancreatic cancer—are hardly detectable with cfDNA in early-stage pancreatic cancer, a low detection rate in our study was inevitable [10].

Moreover, we found that the GSS approach using a small subset of genes, less than 79, could predict OS in both internal and external validation data. Profiling CNAs from only these small sets of genes instead of a genome-wide approach could reduce the test’s cost. Some of these genes were related to pancreatic cancer prognosis. For example, the amplification of KRAS [32] and deletion of CDKN2A [33] are associated with poor prognosis. Elevated expression of CDCP1 [34] has also been reported to be correlated with poor prognosis. Several other genes, such as BCAT1 [35], BCL2 [36], and ATF6 [37], are related to prognosis in other types of cancers. One potential limitation of the external validation of GSS is that the DNA source of the external validation data was not plasma but solid tumor tissue. Even though plasma cfDNA can closely reflect the true CNA profile of tumor tissue, some CNAs could be missed out owing to the low level of ctDNA in plasma cfDNA. In this regard, the statistical significance calculated in external validation might be overestimated.

Focal CNAs correlate with proliferation markers and chromosomal arm level, and whole chromosome CNAs correlate with immune evasion markers [38], implying distinct underlying mechanisms [39]. Current challenges for the clinical application of genomic instability measures are two-fold [40]. The first challenge lies in developing an optimal algorithm to integrate clinical parameters with genomic instability measures, genomic data, and transcriptomic data (germline single-nucleotide polymorphisms, mutations, CNVs, somatic mutations, CNAs, and gene expression differences). The second challenge is the translation of these algorithms into clinical protocols. Despite studies demonstrating the potential of integrated genomic instability measures and clinical parameters, we believe that evaluating these clinical trial tools will be critical to improving pancreatic cancer treatment.

## 5. Conclusions

In conclusion, our findings in a relatively large cohort of patients with PDAC with either metastatic or localized pancreatic adenocarcinoma demonstrated the prognostic value of the ctDNA I-score for survival in this malignancy.

## Figures and Tables

**Figure 1 cancers-13-05466-f001:**
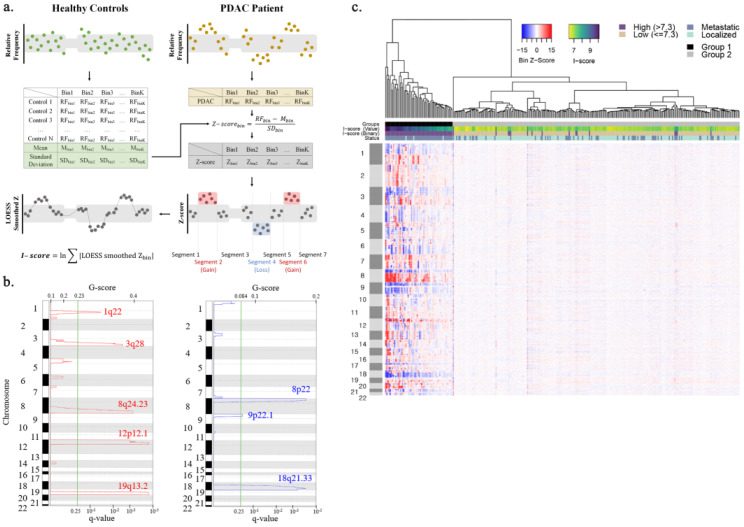
Enriched amplification and deletion regions identified by cfDNA CNA analysis. (**a**) A schematic illustration of the I-score calculation. (**b**) Amplification (red) and deletion (blue) regions identified by GISTIC analysis. G-score and q-value are indicated on the upper and lower side of the x-axis. Cytoband information of each region is written on the y-axis. (**c**) Heatmap showing genome-wide CNV profiles of 315 patients. The top dendrogram shows that 315 patients are well divided into high and low I-score groups. The colored bar right above the heatmap indicates the clustering group, I-score, and tumor status for each patient. The color changes yellow to dark blue as the I-score increases.

**Figure 2 cancers-13-05466-f002:**
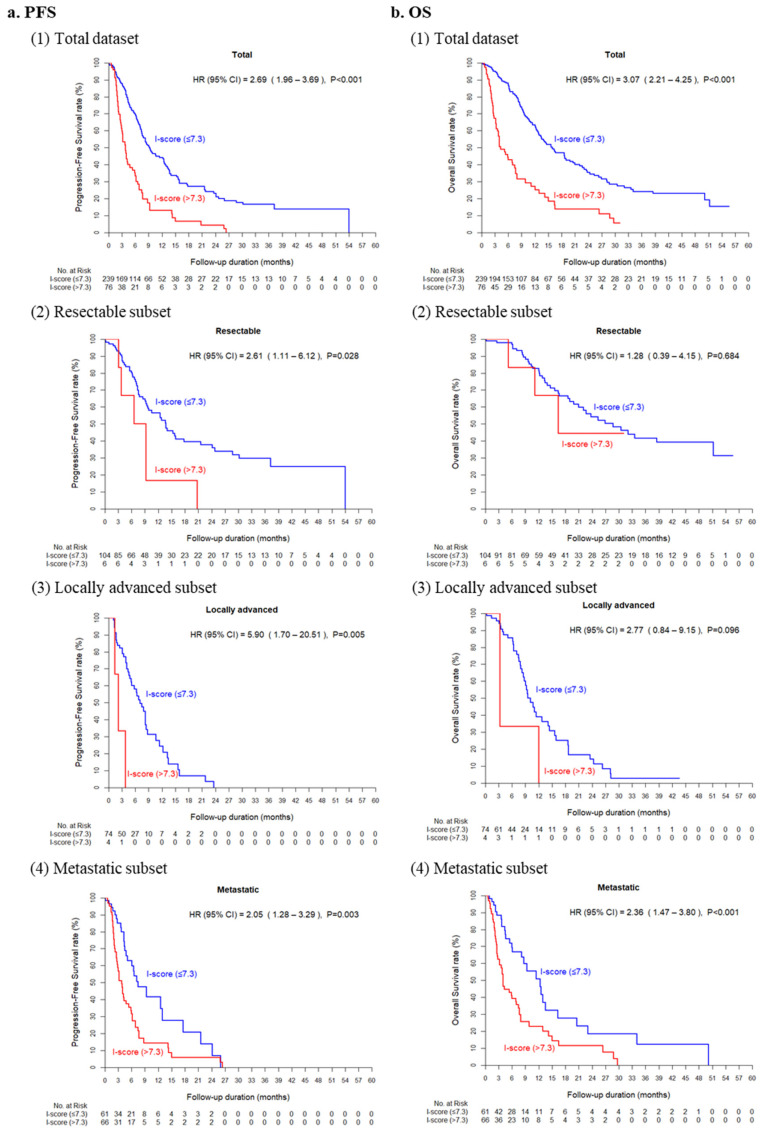
Prognostic impact of circulating tumor DNA I-score. Kaplan–Meier curves for progression-free survival (**a**) and overall survival (**b**) curves according to I-score in total dataset (**1**) and resectable (**2**), locally advanced (**3**), and metastatic (**4**) subsets.

**Figure 3 cancers-13-05466-f003:**
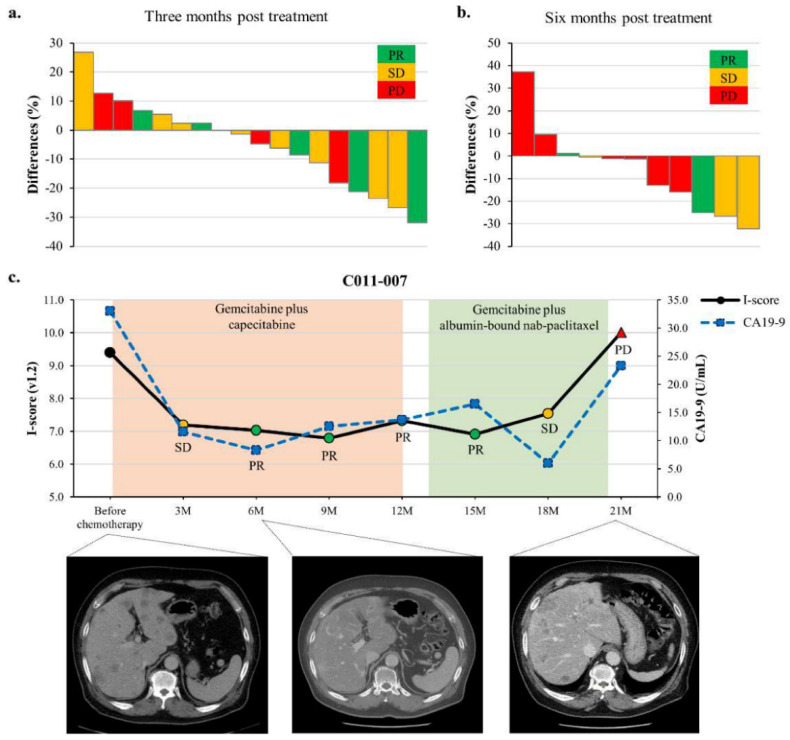
Monitoring of I-score according to responses in patients receiving chemotherapy. (**a,b**) The responses according to changes in I-score in 18 metastatic patients. At six months post-treatment, two patients with 37.2% or 9.6% increased I-score showed progressive disease (PD). In contrast, two (40%) of five patients with a 13.1% or 15.9% decrease in I-score showed progressive disease. (**c**) Case presentation with computerized tomography (CT) images.

**Table 1 cancers-13-05466-t001:** Clinico-pathological characteristics of 315 pancreatic cancer patients stratified by I-score (high vs. low).

Characteristic		Total	I-Score		*p*-Value
		(*n* = 315)	Low (*n* = 239)	High (*n* = 76)	
Age	Mean ± SD	65.4 ± 9.7	65.8 ± 9.6	64.1 ± 10	0.1862
Sex					
	Male	181 (57.5%)	136 (75.1%)	45 (24.9%)	0.7231
	Female	134 (42.5%)	103 (76.9%)	31 (23.1%)	
Status					
	Resectable	110 (34.9%)	104 (94.5%)	6 (5.5%)	<0.0001
	Locally advanced	78 (24.8%)	74 (94.9%)	4 (5.1%)	
	Metastatic	127 (40.3%)	61 (48.0%)	66 (52.0%)	
Pancreatic tumor location					
	Body or tail	164 (52.1%)	108 (65.9%)	56 (34.1%)	<0.0001
	Head or neck	151 (47.9%)	131 (86.8%)	20 (13.2%)	
CA19-9, U/mL ^a^					
	≤37	106 (34.1%)	95 (98.6%)	11 (10.4%)	<0.0001
	>37	205 (65.9%)	141 (68.8%)	64 (31.2%)	
CEA, U/mL ^b^					
	≤5	152 (53.3%)	138 (90.8%)	14 (9.2%)	<0.0001
	>5	133 (46.7%)	77 (57.9%)	56 (42.1%)	
ECOG					
	0 + 1	309 (98.1%)	234 (75.7%)	75 (24.3%)	>0.9999
	2 + 3	6 (1.9%)	5 (83.3%)	1 (16.7%)	

Missing value a = 4, b = 30. Standard deviation, SD; carbohydrate antigen 19-9, CA19-9; carcinoembryonic antigen, CEA; Eastern Cooperative Oncology Group, ECOG.

**Table 2 cancers-13-05466-t002:** Univariable and multivariable analyses of OS and PFS for I-score and clinical factors.

Characteristic		PFS (*n* = 315, Event = 186)				OS (*n* = 315, Event = 167)			
		Univariable Model		Multivariable Model		Univariable Model		Multivariable Model	
		HR (95% CI)	*p*-Value	HR (95% CI)	*p*-Value	HR (95% CI)	*p*-Value	HR (95% CI)	*p*-Value
I-score	Low (≤7.3)	1 (ref)		1 (ref)		1 (ref)		1 (ref)	
	High (>7.3)	2.69 (1.96–3.69)	<0.0001	1.99 (1.42–2.77)	<0.0001	3.07 (2.21–4.25)	<0.0001	2.15 (1.53–3.01)	<0.0001
Age		1.00 (0.98–1.01)	0.674			1.01 (0.99–1.02)	0.5036		
Sex	Male	1 (ref)				1 (ref)			
	Female	0.91 (0.68–1.21)	0.5081			0.76 (0.55–1.04)	0.0842		
Status	Resectable	1 (ref)		1 (ref)		1 (ref)		1 (ref)	
	Locally advanced + Metastatic	2.76 (1.99–3.84)	<0.0001	2.26 (1.60–3.20)	<0.0001	4.05 (2.82–5.82)	<0.0001	3.41 (2.34–4.96)	<0.0001
Pancreatic tumor location	Body or tail	1 (ref)				1 (ref)			
	Head or neck	0.86 (0.65–1.15)	0.3198			0.72 (0.53–0.97)	0.0331		
CA19-9, U/mL ^a^	≤37	1 (ref)				1 (ref)			
	>37	1.69 (1.22–2.33)	0.0014			1.60 (1.14–2.24)	0.0066		
CEA, U/mL ^b^	≤5	1 (ref)				1 (ref)			
	>5	1.74 (1.28–2.35)	0.0004			2.09 (1.51–2.88)	<0.0001		
ECOG	0 + 1	1 (ref)				1 (ref)			
	2 + 3	1.19 (0.49–2.89)	0.7082			0.93 (0.30–2.93)	0.9058		

Missing value a = 4, b = 30. hazard ratio, HR; confidence interval, CI; ref, reference group; progression-free survival, PFS; overall survival, OS; Eastern Cooperative Oncology Group, ECOG; carbohydrate antigen 19-9, CA19-9; carcinoembryonic antigen, CEA.

**Table 3 cancers-13-05466-t003:** Genomic coordinates of GISTIC regions containing known pancreatic cancer related genes.

Amp/Del	GISTIC Amp/Del Region			Curated Pancreatic Cancer-Related Gene				
	Chr	Start	End	Chr	Start	End	Gene	Type
Amp	chr3	174,000,002	198,022,430	chr3	187,439,164	187,463,513	*BCL6*	OCG
				chr3	178,866,310	178,952,497	*PIK3CA*	OCG
	chr8	125,000,002	146,364,022	chr8	128,748,314	128,753,680	*MYC*	OCG
				chr8	141,668,480	142,011,412	*PTK2*	PPA
	chr12	24,000,002	27,000,000	chr12	25,358,179	25,403,854	*KRAS*	OCG
	chr19	31,000,002	59,128,983	chr19	45,251,977	45,263,301	*BCL3*	OCG
				chr19	45,281,125	45,303,903	*CBLC*	OCG
				chr19	40,736,223	40,791,302	*AKT2*	OCG
				chr19	39,390,339	39,399,534	*NFKBIB*	PPA
				chr19	39,078,280	39,108,643	*MAP4K1*	PPA
				chr19	39,876,269	39,881,835	*PAF1*	PPA
				chr19	38,924,339	39,078,204	*RYR1*	PPA
Del	chr9	1	27,000,000	chr9	4,985,244	5,128,183	*JAK2*	OCG
				chr9	21,802,634	21,865,969	*MTAP*	TSG
				chr9	21,967,750	21,994,490	*CDKN2A*	TSG
				chr9	22,002,901	22,009,312	*CDKN2B*	TSG
				chr9	8,314,245	10,612,723	*PTPRD*	TSG

Amplification, Amp; deletion, Del; Oncogene, OCG; tumor suppressor gene, TSG; poor prognosis with amplification, PPA.

## Data Availability

The data presented in this study are available on request from the corresponding author.

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
