# Peer review of "Genomic Instability of Circulating Tumor DNA as a Prognostic Marker for Pancreatic Cancer Survival: A Prospective Cohort Study"

_cancers, 2021, doi:10.3390/cancers13215466_

Round 1
Reviewer 1 Report
This paper is evaluating the use of genetic instability of circulating tumor DNA as a prognostic predictor for pancreatic cancer. As such, this is most likely the very first report of its kind and hence a valuable addition to the scientific knowledge in this field. The paper has been written in logical manner and the work has been conducted correctly as far as I can see. However, there are some aspects which need more focus and amendments, mainly textual and graphical presentation.
Line 29: the English in this sentence should be revised, i.e. “significant factors” should be replaced with “a significant factor”
Line 58: Genomic instability is NOT a hallmark of ALL cancers. Thus, the sentence should be edited to say “hallmark of most cancers” or “one of the typical cancer hallmarks”
Line 70: A couple of additional papers should be added here:
Wang ZY, Ding XQ, Zhu H, Wang RX, Pan XR, Tong JH. KRAS Mutant Allele Fraction in Circulating Cell-Free DNA Correlates With Clinical Stage in Pancreatic Cancer Patients. Front Oncol. 2019;9:1295. Published 2019 Nov 29. doi:10.3389/fonc.2019.01295
Bernard V, Kim DU, San Lucas FA, Castillo J, Allenson K, Mulu FC, Stephens BM, Huang J, Semaan A, Guerrero PA, Kamyabi N, Zhao J, Hurd MW, Koay EJ, Taniguchi CM, Herman JM, Javle M, Wolff R, Katz M, Varadhachary G, Maitra A, Alvarez HA. Circulating Nucleic Acids Are Associated With Outcomes of Patients With Pancreatic Cancer. Gastroenterology. 2019 Jan;156(1):108-118
These are in fact already used elsewhere in this paper: references (27) and (30)
Line 72: “Structural variation” need bit more information here. Is it plain genomic, or genetic as well, in contrast to protein level variation? Or all of them?
Line 76: Use “have been identified” instead of “have been known”
Line 86: Phrase “optimize outcome prediction” should be rewritten with bit more detail or reworded, for example “making more effective” or something similar
Line 97: “and patients”, replace with “and the participating patients gave their informed consent”
Line 102: Define “FOLFIRINOX” here as mentioned for the first time. Definition can hence be removed from Figure 3a explanations.
Line 106-107: Also give RECIST abbreviation here and add reference to the Eisenhauer et al paper from 2009 in European Journal Of Cancer.
Line 114: What was the minimum required/used, is it known?
Line 132: Tapestation4200 should be cited as “Tapestation 4200”. The Agilent Tapestation size distribution data should be given as supplementary information.
Line 158: The copy I was provided had only B&W version of this figure and the panel C needs to be in color to be readable at all. Furthermore, panel b. is way too small so that the cytobands could be read with confidence. Panel b also needs units on horizontal axis.
Line 271: Figure 2 needs improved presentation. If color cannot be used, instead of having the I-score explanation in a box, they could be in close vicinity of the actual curves for easy interpretation.
Line 286: Figure 3, panel a is not actually a figure but a table. The panel a should be renamed as Table 3 and Figure 3 should only have three panels (b, c and d renamed as a, b and c).
Line 308: Due to the long stretches of numbers this table is not easy to read in the present form. The four amplification regions should be marked more clearly, for example “Region 1”, “Region 2” etc. or with background shading. CDKN2A/CDKN2B are also known TSG’s (P14ARF/P15) so not entirely clear why different type indication was used for these, versus JAK2. If there is a reason for this, is should be indicated more clearly.
Line 331.Remove “in a cohort of 315 patients” and add it to the beginning of the sentence, like “this study of 315 patients is the most…”
Reviewer 2 Report
The authors presented a work focused on the role of genomic instability detected at ctDNA levels in PDAC. Despite the project has an average scientific soundness, there are no novelties: the formula of genomic instability is not original and not well described within the work. Materials and methods and Results sections are not well described and organized, with no connections between them.
Reviewer 3 Report
This study investigated genomic instability of circulating tumor DNA (ctDNA) as a prognostic biomarker for pancreatic ductal adenocarcinoma (PDAC). Genomic instability in ctDNA were converted into I-score to segregate patients into low and high genomic instability groups. Overall, it is an important research with some interesting findings. However, there are several issues listed below need to be addressed.
- The paper was written in technical language rather difficult and somewhat inaccessible to non-specialist reader.
- Sequencing was performed to compare ctDNA with matched tumour in 15 resectable pancreatic cancers, only one patient showed concordance. No evidence was provided to demonstrate that ctDNA in patients with non-resectable tumours originated from the tumour.
- It is not clear if different chemotherapeutic treatments can differentially alter the patterns of ctDNA genomic instability in different patient groups? Were patients with metastatic tumours treated with more aggressive drugs or higher drug doses compared to patients with resectable tumours? If so, could the differential drug treatments result in low and high I-scores? If this is the case, differential drug treatment would be a confounding factor. Evidence is needed to rule out the possibility that drug treatment could change the patterns of genomic instability in ctDNA.
Reviewer 4 Report
Dear Authors,
I have read with interest the manuscript from San Myung Woo et al, on the role of circulating tumor DNA as prognostic marker for pancreatic cancer (PC) survival. Briefly, as genomic instability is pivotal in cancer development and progress, the authors assessed the role of ctDNA in predicting PC survival. Of note, ctDNA is a more accessible and collectable material compared to biopsy samples and might act as a proxy for genomic instability.
Thus, in this prospective study the authors evaluated ctDNA in patients with resectable/locally advanced/metastatic PC assessing long-term outcomes in terms of overall survival and progression free survival. To quantified the extent of genome wide copy number instability the authors developed a novel score, the I score. The authors found out that higher I scores were associated with higher HR for both PFS ans OS. In addition, the authors showed that an increase of I score was associated with disease relapse in one patient. Finally, the authors identified a signature of 8 genes that summarizes the genomic instability and predicted best OS.
I have few questions:
- how were the 38 healthy control for the ZEta score selected?
- how was the I score cut off set? is there any possibility to validate the I score?
- could the authors exclude that stage and I score are colinear (for the regression analysis purposes)?
- paragraph 3.3 line 278-280 are not clear. in general, it is not clear which data support the title (ct dna score is associated with response to chemotherapy)
- cfDNA instead of ctDNA isoften used erroneously
- it is not clear which genes from the GISTIC analysis were included into the GSS score and how these genes had been selected
Round 2
Reviewer 2 Report
The work present a lack of novelty.
Reviewer 3 Report
Accept